

# Urbanization-induced urban heat island and aerosol effects on
# climate extremes in the Yangtze River Delta Region of China
Shi Zhong[2, 1], Yun Qian[1*], Chun Zhao[1], Ruby Leung[1], Hailong Wang[1], Ben Yang[3, 1], Jiwen Fan[1],
Huiping Yan[4, 1], Xiu-Qun Yang[3], and Dongqing Liu[5]
[1] Pacific Northwest National Laboratory, Richland, WA, USA
[2] State Key Laboratory of Hydrology-Water Resources and Hydraulic Engineering, Center for
Global Change and Water Cycle, Hohai University, Nanjing, China
[3] School of Atmospheric Sciences, Nanjing University, Nanjing, China
[4] College of Atmospheric Science, Nanjing University of Information & Technology, Nanjing,
China
[5] Nanjing Meteorological Bureau, Nanjing, China
Corresponding author: Yun Qian [Yun.Qian@pnnl.gov]
To be submitted to *Atmospheric Chemistry and Physics*
October 1, 2016





**Abstract**

21        The WRF-Chem model coupled with a single-layer Urban Canopy Model (UCM) is

integrated for 5 years at convection-permitting scale to investigate the individual and combined
impacts of urbanization-induced changes in land cover and pollutants emission on regional
climate in the Yangtze River Delta (YRD) region in eastern China. Simulations with the
urbanization effects reasonably reproduced the observed features of temperature and
precipitation in the YRD region. Urbanization over the YRD induces an Urban Heat Island (UHI)
effect, which increases the surface temperature by 0.53 °C in summer and increases the annual
heat wave days at a rate of 3.7 d/yr in the major megacities in the YRD, accompanied by
intensified heat stress. In winter, the near-surface air temperature increases by approximately 0.7
°C over commercial areas in the cities but decreases in the surrounding areas. Radiative effects
of aerosols tend to cool the surface air by reducing net shortwave radiation at the surface.
Compared to the more localized UHI effect, aerosol effects on solar radiation and temperature
influence a much larger area, especially downwind of the city-cluster in the YRD.

Results also show that the UHI increases the frequency of extreme summer precipitation

by strengthening the convergence and updrafts over urbanized areas in the afternoon, which
favor the development of deep convection. In contrast, the radiative forcing of aerosols results in
a surface cooling and upper atmospheric heating, which enhances atmospheric stability and
suppresses convection. The combined effects of the UHI and aerosols on precipitation depend on
synoptic conditions. Two rainfall events under two typical but different synoptic weather patterns
are further analyzed and the results suggest that synoptic forcing plays a significant role in
modulating the urbanization-induced land-cover and aerosol effects on individual rainfall event.

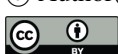



Hence precipitation changes due to urbanization effects may offset each other under different
synoptic conditions, resulting in little changes in mean precipitation at longer time scales.



## 1. Introduction

Urbanization affects climate and hydrological cycle by changing land cover and surface albedo, which releases additional heat to the atmosphere, and by emitting air pollutants, which interact with clouds and radiation (e.g., Shepherd, 2005; Sen Roy and Yuan, 2009; Yang et al., 2011). The most discernible impact of urban land-use change is the urban heat island (UHI) effect that can result in a warmer environment over urban areas than the surrounding areas (Landsberg, 1981; Oke, 1987). In addition to the thermal perturbations, the UHI has been well documented to modify wind patterns (Hjemfelt, 1982), evaporation (Wienert and Kuttler, 2005), atmospheric circulations (Shepherd and Burian, 2003; Baik et al., 2007; Lei et al., 2008), and precipitation around urban areas (Braham, 1979; Inoue and Kimura, 2004). Previous studies have found an increase of warm-season precipitation over and downwind of major cities due to the expanded urban land cover (Huff and Changnon, 1972; Changnon, 1979; Zhong et al., 2015). Recent studies suggested that the underlying urban surface also affects the initiation and propagation of storms (Bornstein and Lin, 2000; Guo et al., 2006) and convective activities in city fringes (Baik et al., 2007; Shepherd et al., 2010).

Concurrently increases in population and anthropogenic activities over urbanized areas increase pollutant emissions and aerosol loading in the atmosphere. Atmospheric aerosols have long been recognized to affect surface and top of the atmosphere (TOA) radiative fluxes and radiative heating profiles in the atmosphere via aerosol-radiation interactions (ARI) (e.g., Coakley et al., 1987; Charlson et al., 1992; Hansen et al., 1997; Yu et al., 2006; Qian et al., 2006, 2007, 2015; McFarquhar and Wang, 2006), which tend to induce cooling near the surface and heating at the low and mid-troposphere (Qian et al., 2006; Bauer and Mennon, 2012). Anthropogenic aerosols can also affect clouds and precipitation via aerosol-cloud interactions





(ACI) (e.g., Rosenfeld, 2000, 2008; Qian et al., 2010; Fan et al., 2013; 2015; Tao et al., 2012;
Zhong et al., 2015). Localized changes in precipitation by strong aerosol perturbations can
induce cold pools by evaporation, which may alter the organization of stratocumulus clouds (e.g.,
Wang and Feingold, 2009; Feingold et al., 2010). Aerosol impacts on deep convective clouds are
complicated by the interactions among dynamical, thermodynamical, and microphysical
processes. For example, deep convection could be invigorated by aerosols as more cloud water
associated with the smaller cloud drops is carried to higher levels where it freezes and releases
more latent heat in a polluted environment (Rosenfeld, 2008; Khain, 2009; Storer and van den
Heever, 2013). Fan et al. (2013) revealed a microphysical effect of aerosols from reduced fall
velocity of ice particles that explains the commonly observed increases in cloud top height and
cloud cover in polluted environments. Therefore, urbanization may influence precipitation and
circulation through multiple pathways that are more difficult to disentangle than the dominant
effect on temperature.

As one of the most developed regions in China, the Yangtze River Delta (YRD) has been

experiencing rapid economic growth and intensive urbanization process during the past three
decades. With the highest city density and urbanization level in China, the YRD has become the
largest adjacent metropolitan areas in the world. It covers an area of $9.96 \times 10^4$ km$^2$, with a total
urban area of $4.19 \times 10^3$ km$^2$ (Hu et al., 2009). Observations have shown that the urban land-use
expansion in this region has induced a remarkable warming due to the significant UHI effect (Du
et al., 2006; Wu and Yang 2012, Wang et al., 2015). The annual mean warming reached up to
0.16°C/10yr based on station measurements in large cities (Ren et al., 2008), which accounted
for 47.1% of the overall warming during the period of 1961-2000. Urbanization in the YRD was
found to destabilize the atmospheric boundary layer (Zhang et al., 2010) and enhance convection





and precipitation (Yang et al., 2012, Wan et al., 2013). Meanwhile, human activities associated
with the ever-growing population have led to a dramatic increase in air pollutant emissions
(Wang et al., 2006). Several observational and numerical studies have revealed that additional
aerosol loading in this region could reduce solar radiation reaching the surface (Che et al., 2005;
Qian et al., 2006, 2007), modify warm cloud properties (Jiang et al., 2013), and suppress light
rainfall events (Qian et al., 2009).
The individual effects of urbanization-induced UHI and aerosol emission on local and
regional climate have been examined separately in several modeling studies using short
simulations of selected weather episodes at high spatial resolution or multiple-year climate
simulations at coarse resolution. To more robustly quantify the urbanization-induced UHI and
aerosol effects, convection-permitting simulations may reduce uncertainties in representing
convection and its interactions with aerosols, which are parameterized in coarse-resolution
models. Additionally, multi-year simulations are needed to understand and quantify the overall
effects of land-cover change and aerosols in different large-scale environments (Oleson et al.,
2008). In this study, a state-of-the-art regional model coupled with online chemistry (WRF-
Chem) and a single-layer Urban Canopy Model (UCM) is used to simulate climate features in
the YRD region. The climatic effects of the separate and combined land-cover and aerosol
changes induced by urbanization are investigated using a set of 5-year (2006-2010) simulations
with a horizontal resolution at convection-permitting scale (3 km). The paper is organized as
follows. Section 2 describes the model configuration, experiment design, and model evaluation.
The urbanization effects on extreme temperature and precipitation are presented in Section 3,
followed by a summary of the conclusions in Section 4.





## 2. Method

### 2.1 Model configuration

The WRF-Chem model (Grell et al., 2005; Fast et al., 2006; Qian et al., 2010) simulates trace gases, aerosols and meteorological fields interactively (Skamarock et al., 2008; Wang et al., 2009), including aerosol-radiation interactions (Zhao et al., 2011, 2013a) and aerosol-cloud interactions (Gustafson et al., 2004). The coupled single-layer UCM (Kusaka et al., 2001; Chen et al., 2001) is a column model that uses a simplified geometry with two-dimensional, symmetrical street canyons to represent the momentum and energy exchanges between the urban surface and the atmosphere. The RADM2 (Regional Acid Deposition Model 2) gas chemical mechanism (Stockwell et al., 1990) and the MADE (Modal Aerosol Dynamics Model for Europe) and SORGAM (Secondary Organic Aerosol Model) aerosol module (Schell et al., 2001) are used. Detailed configuration of the above models can be found in Zhao et al. (2010). No cumulus parameterization is used at the convection-permitting resolution. The physical parameterization schemes used in our simulations are listed in Table 1.

### 2.2 Numerical experiments

Simulations are performed over a model domain centered at (120.50 °E, 31.00 °N) with a horizontal grid spacing of 3 km and 50 vertical levels extending from the surface to 50 hPa. The lowest 10 model layers are placed below 1 km to ensure a fine vertical resolution within the planetary boundary layer. Initial and boundary conditions for meteorological fields are derived from the National Center for Environmental Prediction (NCEP) FNL global reanalysis data on 1° × 1° grids at 6-hour interval. Lateral boundary conditions for chemistry are provided by a quasi-





global WRF-Chem simulation (Zhao et al., 2013b) that includes aerosols transported from
regions outside the model domain.
The dominant land cover within each model grid cell is derived from the U.S. Geological
Survey (USGS) 30 second dataset that includes 24-category land-use type, except that the land
use over urban areas is updated using the stable nighttime light product (version 4) at 1 km
spatial     resolution     (available     at     the     National     Geophysical     Data     Center,
http://ngdc.noaa.gov/eog//dmsp/downloadV4composites.html). Corresponding to the value of
lighting index of 25-50, 50-58, and >58 in the above product, each urban grid is identified as
"Low    Intensity    Residential    (LIR)",    "High    Intensity    Residential    (HIR)",    or
"Commercial/Industrial/Transportation (CIT)", respectively. Figures 1a and 1b illustrate the
urban area within the model domain for year 1970 and 2006, respectively. The anthropogenic
heating (AH), characterized by a diurnal cycle with two peaks at rush hours of 0800 and 1700
LST, respectively, is incorporated in the model simulations. The default maximum values of AH
in WRF for LIR (20 W m$^{-2}$), HIR (50 W m$^{-2}$) and CIT (90 W m$^{-2}$) are used in this study (Tewari
et al., 2007). Anthropogenic emissions of aerosols and their precursors are obtained from the
Asian emission inventory (Zhang et al., 2009b), which is a $0.5° \times 0.5°$ gridded dataset for 2006.
Black carbon (BC), organic matter (OM), and sulfate emissions over China are extracted from
the China emission inventory for 2008 (Lu et al., 2011), which provides monthly mean data on
$0.1° \times 0.1°$ grids. It should be noted that the Noah land surface model defines a dominant land
cover type for each grid, so no subgrid variability is simulated.
The anthropogenic emission fluxes of $SO_2$ and BC in the simulation domain are shown in
Figures 1c and 1d, respectively. Areas with large emissions are mainly located in four city
clusters, i.e., Nanjing-Zhenjiang-Yangzhou, Suzhou-Wuxi-Changzhou, Shanghai, and Hangzhou



Bay, all inside the mega-city belt. Biomass burning emissions for the simulation period are
obtained from the monthly Global Fire Emissions Database Version 3 (GFEDv3), which
provides monthly mean data on $0.5° \times 0.5°$ grids and the vertical distribution is determined by
the injection heights described by Dentener et al. (2006) for the Aerosol Inter-Comparison
project (AeroCom). Sea salt and dust emissions are configured following the same approach of
Zhao et al. (2013b).
In order to investigate the individual responses of local and regional climate to land-cover
change and increased aerosol loading, three experiments (i.e., LU06E06, LU70E70, and
LU70E06) are conducted for 5 years from 2006 to 2010. The configurations of land use and
aerosol emissions for these experiments are summarized in Table 2. All three simulations are
performed using the same initial and boundary conditions and physics schemes, but with
different land use types and/or anthropogenic emissions. LU06E06 is the control experiment,
which represents the "present" (2006) urbanization level for both land use and aerosol/precursor
emissions. LU70E06 uses the present aerosol emission data but with the land use of the 1970s,
which is derived from the USGS dataset without the nighttime light correction. In LU70E70,
both land use and emissions are set to the conditions of the 1970s. The differences of LU06E06-
LU70E06, LU70E06-LU70E70, and LU06E06-LU70E70 can be used to derive the urban land-
use effect, aerosol effect, and their combined effect, respectively (Table 3). The simulations are
initialized on December 15 of each year during 2005-2009 to allow for a 16-day spin-up time
and then continuously integrated for the next year (from January 1 to December 31). Results
from January 1 to December 31 of all five years (2006-2010) are analyzed.
**2.3 Model evaluation**




The surface skin temperature simulated in LU06E06 is averaged over 2006-2010 and
compared with the MODIS data. A spatial filtering method described by Wu and Yang (2012) is
applied to isolate the heterogeneous climatic forcing of urbanization. More specifically, for each
grid a spatial anomaly is defined as the departure from the average value over a region centered
at each grid. Then, the moving spatial anomalies are calculated for all the grids with the moving
region acting as a filtering window, which has a size of 1° ×1°. Figure 2 shows the moving
spatial anomalies of mean surface skin temperature from MODIS observations and the L06E06
simulation. The simulation captures the spatial distribution of observed surface skin temperature
very well. In particular, the warmer centers over highly urbanized areas are well reproduced,
despite slight underestimations in some mega cities in Zhejiang Province such as Hangzhou and
Ningbo. Shanghai and Su-Xi-Chang exhibit the highest temperatures that are 2 °C above the
surrounding rural areas.
To further validate the model, the baseline simulation LU06E06 is evaluated against
meteorological station observations for 2006-2010. Figure 3 shows the averaged near-surface
temperature and precipitation from observations and LU06E06. The simulated spatial pattern of
near-surface air temperature agrees well with observations, with high temperature centers located
at meteorological stations in major cities such as Shanghai and Hangzhou. The simulated
temperature displays substantial spatial variability associated with heterogeneity in topography,
land cover, and other regional forcings. The model captures the general north-to-south gradient
of increasing precipitation in the observations. However, the model overestimates precipitation in
Shanghai and central Jiangsu Province but underestimates the precipitation in the southwestern
part of the domain.



## 3. Results

### 3.1 Urbanization impact on surface temperature, radiation flux and heat waves

#### 3.1.1 Mean near-surface air temperature

Figure 4 shows the differences in 2-meter near surface air temperature (T2m) among the three experiments to quantify the UHI and aerosol effects from urbanization (Table 3). The UHI effect causes an increase in near-surface temperature over the urbanized area in summer. The average temperature increase is about 0.53 °C over urban area and 1.49 °C in commercial areas outlined by the green contours (see Fig. 4a). In winter, the UHI warming effect occurs primarily in commercial areas, where the mean temperature increases by about 0.7 °C. In areas surrounding the central commercial region, however, temperature decreases due to the urban land-cover change (shown in Fig. 4d). Such a cooling effect in winter has also been found in previous studies (e. g., Oke, 1982; Jauregui et al., 1992; Wang et al., 2007). The "cool island" effects of urbanization during daytime in winter can be explained by the much larger surface thermal inertia of urban areas than that of rural areas with very low vegetation cover during winter (Wang et al., 2007). Although the wintertime cooling effect in urbanized area is not widely recognized, it is an important phenomenon that is also simulated by the model.

The increased aerosols induced by urbanization exert a cooling effect over the entire simulation domain in both summer and winter (Fig. 4b and 4e). On a domain average, the temperature reduction induced by increased aerosols is less than the warming induced by the UHI effect in both seasons. Therefore, the net urbanization impact (including both land-cover change and aerosol increase) on near-surface temperature is dominated by the UHI warming effect (Fig. 4c and 4f) resulted from the land-cover change in the YRD.



### 3.1.2 Surface solar radiation

The effects of urban land-cover change and increased aerosols on surface net shortwave
radiation are shown in Fig. 5. As the building clusters reduce surface albedo (Oke, 1987), land-
cover change increases the net shortwave radiation over urbanized areas, with an average
increase of 9.11 W m$^{-2}$ in summer and 8.49 W m$^{-2}$ in winter. The net increase is greater in
summer than in winter because of the stronger summertime incoming solar radiation. On the
contrary, aerosols reduce the surface net shortwave radiation in the northern part of the domain
corresponding to the larger $SO_2$ and BC emission rates (Fig. 1), with a magnitude of 8.79 W m$^{-2}$
in summer and 7.63 W m$^{-2}$ in winter. Different from the UHI effect that is more localized, the
radiative impact of aerosols is more widespread and significant west of the major urban areas
and even over the ocean. Figure 6 shows the spatial pattern of mean surface winds simulated in
LU06E06 and the difference in column-integrated $PM_{2.5}$ mass concentration between LU70E06
and LU70E70. Consistent with the prevailing monsoon circulation, southeasterly (northeasterly)
flows dominate the YRD in summer (winter), which lead to increases in the $PM_{2.5}$ concentration
over the downwind area of the YRD city clusters. The increased $PM_{2.5}$ concentrations downwind
of the YRD reduce solar radiation to the west (southwest) of the YRD in summer (winter), as
shown in Figs. 5b and 5d. Hence aerosol effects on radiation are not limited to the emission
source areas in metropolitan regions.

### 3.1.3 Heat waves

The UHI effect can significantly increase the near-surface temperatures in summer,
thereby exacerbating extreme heat waves in urbanized areas (Stone, 2012). By definition, a heat
wave occurs when the near-surface temperature reaches or exceeds 35 °C for three or more





consecutive days (Tan et al., 2004). The averaged heat wave days comparing LU06E06 and
LU70E06 increase at a rate of 3.7 d/yr in the major mega cities (Fig. 7a). The increase is most
pronounced in Shanghai, with a rate larger than 12 d/yr.

High temperature during heat wave contributes to heat exhaustion or heat stroke, but the

impact of atmospheric humidity on evaporation is also crucial. Here we use a heat stress index to
assess the combined effects of temperature and humidity on human health due to the UHI effect,
expressed as (Masterson and Richardson, 1979):

$$Humidex = Ta + (5/9)(e - 10) \qquad (1)$$

where $Ta$ is near-surface air temperature (°C) and $e$ is water vapor pressure (hPa). Figure 7b
depicts a big increase in heat stress index (Humidex) over urbanized regions in the YRD, except
for the city of Hangzhou. The increase in heat stress index is more accentuated in Shanghai, with
a mean increase of 2.16, relative to other urban areas. This suggests that humidity has a larger
influence on heat stress in Shanghai because of its proximity to the ocean compared to urban
areas further inland. In contrast, increased aerosols have little impact on heat waves (results not
shown) because their impacts on near-surface temperature are much weaker (Fig. 4b).

**3.2 Urbanization effects on summertime precipitation**

**3.2.1 Long-term impact on extreme rainfall**

Previous studies have provided evidence of urbanization effect on precipitation

distribution in and around urban areas (e.g. Shepherd et al., 2003; Kaufmann et al., 2007; Miao et
al., 2010). Several mechanisms have been proposed for the effects of urbanization on
precipitation: (1) the UHI effect can destabilize the planetary boundary layer (PBL) and trigger





convection; (2) increased surface roughness may enhance atmospheric convergence that favors
updrafts; (3) building obstruction tends to bifurcate rainfall systems and delays its propagation;
(4) the change in land-cover decreases local evaporation, (5) anthropogenic emissions increase
aerosol loading in the atmosphere, with subsequent effects on precipitation through changes in
radiation and cloud processes. These mechanisms contribute to positive and negative changes in
precipitation, leading to more complicated effects on precipitation than temperature.

In this section we analyze the results of the three 5-year simulations to examine the long-

term impact of urbanization on precipitation. The results show that influences of both urban land
cover and elevated aerosols on annual and seasonal mean precipitation are relatively small (not
shown). This may be due to the urbanization effect for different rainfall events offsetting each
other, leading to an overall weak effect on a longer time scale (see Section 3.2.2). Here we focus
on the frequency of extreme rainfall over the YRD region. Extreme summer rainfall events are
defined using hourly precipitation rate that is above $95^{th}$ percentile at each grid for the period of
2006-2010. Figure 8 shows the diurnal cycles of extreme rainfall frequency and urbanization-
induced changes in the areas around Nanjing, Shanghai, and Su-Xi-Chang (shown in Fig. 1b).
The frequency of hourly extreme rainfall reaches its maximum at around 16:00-17:00 LST over
three urban clusters. Urban land-cover change increases the occurrence of extreme precipitation
in the afternoon (12:00 to 20:00 LST). The maximum increase in the frequency of extreme
hourly rainfall events for Nanjing, Shanghai, and Su-Xi-Chang can reach 0.86%, 1.09%, and
0.79%, respectively, with the peak increase occurring in the late afternoon. On the contrary,
aerosols exert an opposite impact to substantially reduce the frequency of extreme rainfall in the
afternoon by up to 1.05%, 0.75%, and 0.72% for Nanjing, Shanghai, and Su-Xi-Chang,
respectively. These impacts are significant compared to the maximum frequency of hourly



extreme rainfall of about 10% in each area. However, opposite effects of land-cover and aerosol
emission changes result in a small net urbanization effect on extreme precipitation.
Because urbanization influences extreme precipitation primarily in the afternoon, we further
analyze extreme rainfall events with a focus on the averages from 1200 to 2000 LST. Figure 9
shows the substantial increase in extreme precipitation frequency concentrated over the major
metropolitan areas in the YRD, with some compensation in the surrounding areas in general.
Aerosols, however, reduce the occurrence of extreme precipitation more uniformly in most areas
of the domain. The most significant influence of aerosols is found in the northwest part of the
domain where aerosol concentrations increase the most downwind of the urban centers (Fig. 6a).
Similar to the effects on surface temperature and solar radiation (Figs. 4 and 5), aerosols have a
substantial impact on the occurrence of extreme precipitation over a wider area than the effects
of urban land-use change.
How do changes in land cover and aerosols modulate extreme rainfall frequency? Figure
10a shows the diurnal time-height cross section of the impact of urban land-cover (i.e., the
difference between LU06E06 and LU70E06) on temperature and divergence averaged over the
three city clusters (Nanjing, Shanghai, and Su-Xi-Chang). Air temperature over the urbanized
areas increases significantly in the afternoon (from 1200 to 1800 LST) due to the UHI effect. The
warming and the increased roughness length in urban areas favor convergence in the lower
atmosphere and divergence above. As a result, the mean updraft increases over the urbanized
areas in the afternoon (Fig. 10b), which increases cloud water from the lower to middle
troposphere in the afternoon. Shortly before noon, there is a small reduction in low clouds, which
may be related to the reduced relative humidity due to warmer temperature and/or reduced
evaporation from the urban land cover, the so-called urban dry island effect (e.g., Hage, 1975;





Wang and Gong, 2009). The increase in cloud water in the afternoon is consistent with the
enhanced updrafts. This mechanism potentially explains the increased frequency of extreme
precipitation in urban areas in the afternoon (e.g. Craig and Bornstein, 2002; Rozoff et al., 2003;
Wan et al., 2013; Zhong and Yang, 2015a, 2015b).

To understand the aerosol-induced reduction in extreme rainfall events, we analyze the

diurnal cycle of aerosol effect (i.e., the difference LU70E06 and LU70E70) on radiative heating,
vertical velocity, and net solar radiation at the surface (Fig. 11). As BC emission rates are
relatively high in the YRD region (Fig. 2d), aerosols heat the atmosphere due to absorption of
solar radiation during daytime (from 08:00 to 17:00 LST). As a result of absorption and
scattering of solar radiation by aerosols, less solar radiation reaches the surface. These changes at
the surface and in the atmosphere stabilize the atmosphere and reduce convective intensity in the
afternoon (from 14:00 to 20:00 LST), which reduces the frequency of extreme rainfall events
(Koren et al., 2004; Qian et al., 2006; Zhao et al., 2006; 2011; Fan et al., 2007). Although
aerosols can enhance precipitation through cloud microphysical changes that invigorate
convection (e.g., Khain et al., 2009; Rosenfeld et al., 2008; Fan et al., 2013), aerosol radiative
effects generally dominate in China because of the high AOD and strong light-absorbing aerosol
properties (Yang et al., 2011; Fan et al., 2015).

### 3.2.2 Synoptic influence on urbanization impacts

The impacts of urbanization-induced UHI and aerosols on precipitation may be highly

variable under different synoptic conditions that influence the atmospheric circulation and cloud
and boundary layer processes. Precipitation changes due to urbanization effects may offset each
other under different synoptic conditions, leading to an overall weak effect on mean precipitation





at longer time scales as discussed in section 3.2.1. We select two typical heavy late-afternoon
rainfall events with different background circulations over the YRD region. Case A occurred
from 08:00 LST 23 June to 08:00 LST 24 June 2006 and case B occurred from 08:00 LST 1 July
to 08:00 LST 2 July 2006. Figure 12a and 12d show the mean precipitation rate and 850 hPa
winds for case A and case B, respectively. Southwesterly flow dominates the entire region in case
A (Fig. 12a), while in case B (Fig. 12d) southwesterly and northwesterly winds dominate the
southern and northern parts of precipitation area, respectively. The averaged background wind
speed in case B is much stronger than that in case A, representing stronger synoptic forcing in
case B. The effects of urban land-cover change and aerosols on precipitation for the case A (case
B) are illustrated in Figs. 12b and 12c (Figs. 12e and 12f), respectively. Both cases show
significant precipitation responses to the forcing of urban land-cover and aerosols. We can see
that urban land cover increases the rainfall intensity in case A but aerosols decrease precipitation
over the urbanized area (Figs. 12b and 12c). The precipitation response to urban land cover and
aerosols is just the opposite in case B (Figs. 12e and 12f). Figs. 13a and 13d illustrate the
evolution of precipitation in region R1 (Fig. 12a) and R2 (Fig. 12d), respectively, for the two
cases. In both cases, rainfall mainly occurred between 08:00 LST and 20:00 LST. The
corresponding impacts of urban land-cover and aerosols are shown in Figs. 13b-c and Figs. 13e-f
for cases A and B, respectively. In case A, the urban land-cover substantially increases the
precipitation intensity in the afternoon with a maximum increase of 6.87 mm h$^{-1}$. Aerosol effects,
on the contrary, decrease the rainfall intensity with a maximum reduction of 3.85 mm h$^{-1}$. In case
B, however, effects of urban land-cover and enhanced aerosols on precipitation are opposite to
that in case A. A maximum rainfall reduction of 3.81 mm h$^{-1}$ is found to be associated with the
effect of urban land cover and an increase of 2.85 mm h$^{-1}$ is associated with the aerosol forcing.





Why do urban land-cover and aerosols exert opposite effects on precipitation during the

two rainfall events? Here we attempt to answer this question by examining the dynamical and
thermodynamical changes induced by the UHI and aerosols using the moisture flux convergence
(MFC), which is defined as:
$$\text{MFC} = -\nabla \cdot \left( q\overrightarrow{V_h} \right) = -q\nabla \cdot \overrightarrow{V_h} - \overrightarrow{V_h} \cdot \nabla q \qquad (2)$$

The first and second terms on the right hand side of Eq. 2 denote wind convergence (CON) and
moisture advection (MA), respectively.

Figures 14a and 14b illustrate the time-height cross sections of changes in moisture flux

convergence and cloud water mixing ratio induced by land-cover and aerosol changes over the
region R1 (Fig. 12a) during the rainy period in case A. Urban land-cover enhances the
convergence of moisture fluxes in the lower troposphere, which results in increased precipitation
(Fig. 14a). On the contrary, aerosols weaken the convergence of moisture fluxes and thus reduce
precipitation (Fig. 14b). These changes are consistent with those associated with extreme rainfall
changes shown in Fig. 10. Interestingly for case B over R2, urban land-cover weakens the
convergence of moisture fluxes (Fig. 14c) and thus suppresses precipitation (Fig. 13e) from
08:00 LST 1 July to 02:00 LST 2 July 2006. Aerosols, however, enhance the convergence of
moisture fluxes over R2 (Fig. 14d) and thus increase precipitation (Fig. 13f). These results
establish obvious correspondence between moisture flux convergence changes and the
precipitation response to urban land cover and aerosols in the two rainfall events and suggest
different processes may dominate the moisture flux convergence changes for the two cases.

Figure 15 presents the time-height cross section of the changes in the two terms of MFC,

i.e., CON (convergence) and MA (moisture advection), induced by land-cover and aerosol





changes averaged over R1 (Fig. 12a) for case A and over R2 (Fig. 12d) for case B. Urban land-
cover enhances the wind convergence over R1 in case A (Fig. 15a), leading to an increase in
CON by up to $1.56 \times 10^{-4}$ g kg$^{-1}$ s$^{-1}$, which is much larger than the increase of $0.61 \times 10^{-4}$ g kg$^{-1}$ s$^{-1}$
averaged over R2 (Fig. 15c) in case B. The larger enhancement of convergence in case A is
attributed to the strong UHI-induced surface heating during this rainfall period (figure not
shown). In contrast, aerosols reduce the convergence in both case A and case B due to the aerosol
cooling effect near the surface, as discussed previously (Fig. 11). The reduction of convergence
in case A is more significant than that in case B because of the larger aerosol loading and,
therefore, stronger surface cooling over R1 in case A (not shown). Urban land-cover reduces
moisture advection in both cases, with a maximum decrease of -0.99 and -1.89 $10^{-4}$ g kg$^{-1}$ s$^{-1}$,
respectively. Aerosols, however, increase moisture advection, and the maximum increases are
0.93 and 1.31 $10^{-4}$ g kg$^{-1}$ s$^{-1}$ in case A and case B, respectively. Our results show clearly that the
changes in CON are opposite to that in MA. As the impacts of urban land-cover and aerosols on
moisture advection are greater in case B than in case A, the net changes in the moisture flux
convergence are dominated by MA in case B and by CON in case A, leading to opposite effects
between the two cases.

The significant differences in the responses of MA between the two cases are related to

different background circulations during the two events (Figs. 12a and 12d). Weaker
southwesterly flow dominates the entire region in case A (Fig. 12a), while in case B (Fig. 12d)
stronger southwesterly and northwesterly winds dominate the southern and northern parts of
precipitation area, respectively. Figure 16 illustrates the time-height cross-section of changes in
wind speed and moisture flux induced by urban land cover and aerosols over R1 for case A and
over R2 for case B. Wind speed in the lower troposphere decreases due to the UHI effect and





increases due to aerosol effects in case A. Corresponding to the changes in wind speed, the water
vapor flux is reduced by the UHI effect and increased by aerosols. These changes are much
larger and extend higher in altitude in case B because of the stronger background winds.

In summary, case B represents stronger synoptic forcing than case A. The stronger winds

and larger spatial coverage of clouds and precipitation associated with the larger scale synoptic
system weakens the UHI and aerosol effects through ventilation and changes in radiation,
resulting in weaker CON and larger MA changes. Conversely, with weaker synoptic forcing, the
stronger UHI and aerosol effects enhance the changes in CON while MA effects are smaller due
to the weaker background winds. Therefore, our results highlight the distinguishing role of
synoptic forcing on how urban land-cover and aerosol influence the dynamical and thermo-
dynamical environments and precipitation.

## 421    4. Summary

In this study, the state-of-the-art WRF-Chem model coupled with a single-layer UCM, is

run at convection-permitting scale to investigate the influences of urbanization-induced land-
cover change and elevated aerosol concentrations on local and regional climate in the Yangtze
River Delta (YRD) in China. A 5-year period (2006-2010) is selected for multi-year simulations
to investigate urbanization effects on extreme events and the role of synoptic forcing. Three
experiments were conducted with different configurations of land cover and aerosol emissions:
(1) urban land and emissions in 2006, (2) urban land in the 1970s and emissions in 2006, and (3)
urban land and emissions in the 1970s. The experiment with the 2006 land-use type and
anthropogenic emissions reproduces the observed spatial patterns of near-surface air temperature
and precipitation fairly well.





The expanded urban land cover and increased aerosols have opposite impacts on the near-
surface air temperature. The urban land-use change increases 2-m air temperature due to the UHI
effect in commercial areas with a domain-averaged increase of 1.49 °C in summer and 0.7 °C in
winter. In the surrounding areas, however, surface air temperature increases in summer but
decreases in winter. The latter is attributed to the much greater thermal initial over urban areas
than over rural areas in wintertime when both vegetation cover and soil moisture are at their
seasonal minimum. Compared to the effect of land-cover change, aerosol effect exerts a less
significant influence on near-surface temperature with minor decreases in both summer and
winter. Overall, the impact of urban land-use change outweighs that of enhanced aerosols on
regional temperature especially in summer. The increase in near-surface temperature induced by
the UHI effect leads to an increase in heat wave days by 3.7 days per year over the major mega
cities in the YRD region. The greater response of solar radiation to urban land-cover in summer
is the major factor contributing to the larger changes in surface temperature in summer than in
winter. Compared to the urban land-use effect, aerosol effect on reducing the surface solar
radiation occurs over a much broader region including the downwind area of the city clusters.
The urban land-cover change and increased aerosols have opposite effects on the
frequency of extreme rainfall during summer. The UHI effect leads to more frequent extreme
precipitation over the urbanized area in the afternoon because of an enhanced near-surface
convergence and vertical motion. In contrast, aerosol tends to decrease the frequency of extreme
precipitation because of its cooling effect near the surface and heating effect (by light-absorbing
particles) above, leading to an increased atmospheric stability and weakened updrafts. Additional
aerosols can also induce decreases in the frequency of extreme precipitation over non-urban
areas, particularly in the downwind area of the city clusters.



The effects of both urban land-cover and increased aerosols on summertime rainfall vary
with synoptic weather systems and environmental conditions. Two late-afternoon rainfall events
are selected for in-depth analysis. For the two cases, urbanization exerts similar impacts on local-
scale convergence and mean wind speed, which modify the strength of moisture transport. More
specifically, the effect of urban land-cover increases local-scale convergence due to the UHI-
induced circulation and reduces low-level wind speed, while aerosols have an opposite effect due
to the cooling near the surface. We found that the impacts of urban land-cover and aerosol on
precipitation are determined not only by their effect on local-scale convergence, but also
modulated by the large-scale weather systems. Our analyses suggest that synoptic forcing plays a
significant role in how urbanization-induced land-cover and aerosols influence individual rainfall
event. Although the two rainfall events selected for the analysis do not represent all types of
precipitation events in the YRD Region, they demonstrate how the effect of urbanization on
precipitation may vary and offset each other under different synoptic conditions, leading to an
overall weak effect on mean precipitation at longer time scales. To further quantify urbanization
effects, uncertainties in anthropogenic emissions and heating, unresolved urban building and
streets structure, and representation in aerosol-cloud interactions and cloud microphysics in the
model should be investigated in future studies. Further investigation is also needed to have a
better and more comprehensive understanding of the complicated mechanisms through which
urbanization influences heavy rainfall under a full range of weather conditions.

**Acknowledgments**
The contributions of PNNL authors are supported by the U.S. Department of Energy's
Office of Science as part of the Regional and Global Climate Modeling Program and



Atmospheric System Research (ASR) program. The contribution of Shi Zhong and Xiu-Qun
Yang is supported by the National Basic Research Program of China (2010CB428504), Jiangsu
Collaborative Innovation Center for Climate Change, and the Scholarship Award for Excellent
Doctoral Student granted by China Scholarship Council. The work of Ben Yang is supported by
the National Natural Science Foundation of China (41305084). Computations were performed
using resources of the National Energy Research Scientific Computing Center (NERSC) at
Lawrence Berkeley National Laboratory and PNNL Institutional Computing. The Pacific
Northwest National Laboratory is operated for DOE by Battelle Memorial Institute under
contract DE-AC05-76RL01830. All model results are archived on a PNNL cluster and available
upon request. Please contact Yun Qian (yun.qian@pnnl.gov).








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





## Table and Figure Captions

**Table 1** Configurations of the WRF physics schemes used in the present study.

**Table 2** Numerical experiments and corresponding urban land use and aerosol emissions.

**Table 3** Analysis strategies for the investigation of urban land-use and/or aerosol effects.

**Figure 1** Land-use categories for year (a) 1970; (b) 2006; and (c) $SO_2$ (units: mol km$^{-2}$ h$^{-1}$) and (d) black carbon (BC) emission rates (units: ug m$^{-2}$ s$^{-1}$) averaged over 2006-2010. Surface topography is also shown in Fig. 1a (contour; units: m). The boxes in Fig. 1b outline three mega-city clusters of Nanjing, Su-Xi-Chang, and Shanghai.

**Figure 2** Moving spatial anomalies of averaged surface skin temperature (units: °C) with a filtering window size of 1° ×1° for (a) MODIS observation and (b) the L06E06 simulation. The "High Intensity Residential" and "Commercial/Industrial/Transportation" areas are marked with green lines and yellow lines, respectively.

**Figure 3** Annual mean (a) near-surface temperature (units: °C) and (b) precipitation (units: mm d$^{-1}$) from observations (shaded circles) and the LU06E06 simulation (shaded).

**Figure 4** Differences in mean 2-m temperature (Units: °C) between simulations (a, d) LU06E70 and LU70E70, (b, e) LU70E06 and LU70E70, (c, f) LU06E06 and LU70E70 for summer (upper panels) and winter (bottom panels). "Commercial/Industrial/Transportation" areas are marked with green lines. The black dots mark the area with statistically significant changes.



**Figure 5** Differences in net shortwave fluxes at the surface (units: W m$^{-2}$) between simulations
(a, c) LU06E70 and LU70E70, and (b, d) LU70E06 and LU70E70 in summer (upper panels) and
winter (bottom panels).
**Figure 6** Differences in column burden of PM2.5 (g m$^{-2}$) between simulations LU70E06 and
LU70E70, superimposed with near-surface winds simulated in LU70E70, for (a) summer and (b)
winter.
**Figure 7** Differences in mean summertime (a) heat wave days (units: d/yr) and (b) heat stress
(units: °C) between simulations LU06E70 and LU70E70.
**Figure 8** Diurnal cycles of the frequency of summertime extreme rainfall events (defined using
hourly precipitation intensity above 95$^{th}$ percentile, black lines) and the differences between
simulations LU06E70 and LU70E70 (red lines), LU70E06 and LU70E70 (blue lines), and
LU06E06 and LU70E70 (green lines) over (a) Nanjing, (b) Shanghai, and (c) Su-Xi-Chang.
**Figure 9** Differences in the frequency of summertime extreme rainfall events (averaged from
12:00 to 20:00 LST) between simulations (a) LU06E70 and LU70E70, and (b) LU70E06 and
LU70E70.
**Figure 10** (a) Time-height cross-sections of differences (between LU06E70 and LU70E70) in
temperature (contour; units: °C) and divergence (shade; units: 10$^{-5}$ s$^{-1}$) averaged over the three
city clusters (Nanjing, Shanghai, and Su-Xi-Chang); (b) same as (a), but for vertical velocity
(shade; units: 10$^{-2}$ m s$^{-1}$) and cloud water mixing ratio (contour; 10$^{-3}$ kg kg$^{-1}$).
**Figure 11** Time-height cross-sections of differences between LU70E06 and LU70E70 in
radiative heating profile (shade; units: K d$^{-1}$), vertical velocity (contour; units: 10$^{-2}$ m s$^{-1}$) and



surface solar radiation (blue bars; units: W m$^{-2}$) averaged over the three city clusters (Nanjing,
Shanghai, and Su-Xi-Chang).
**Figure 12** Rain rate (units: mm h-1) superimposed with wind vectors at 850 hPa for case A from
08:00 LST 23 June to 08:00 LST 24 June 2006 (a) simulated in the LU06E06 simulation, (b)
differences between LU06E70 and LU70E70, (c) differences between LU70E06 and LU70E70.
Panels (d-f) are the same as (a-c) but for case B from 08:00 LST 1 July to 08:00 LST 2 July
2006**.** The boxes R1 in (a) and R2 in (d) outline the three regions over which further analysis are
conducted. Lines across the center of each box mark the cross-sections to be analyzed.
**Figure 13** The time evolution of precipitation (units: mm h$^{-1}$) along the line *ab* (marked in Fig.
12a) from 08:00 LST 23 June to 02:00 LST 24 June 2006 (case A) (a) simulated in the LU06E06
simulation, (b) differences between LU06E70 and LU70E70, (c) differences between LU70E06
and LU70E70. Panels (d-f) are the same as (a-c) but for case B along line *cd* (marked in Fig.
12d) from 08:00 LST 1 July to 02:00 LST 2 July 2006**.**
**Figure 14** The time-height cross-sections of differences in moisture flux convergence (shaded;
units: $10^{-4}$ g$^{-1}$ kg$^{-1}$ s$^{-1}$) and water vapor mixing ratio (black lines; units: $10^{-2}$ g kg$^{-1}$) from 08:00
LST 23 June to 02:00 LST 24 June 2006 (case A) over region R1 (denoted in Fig. 12a) between
(a) LU06E70 and LU70E70; (b) LU70E06 and LU70E70; Panels (c, d) are the same as (a, b) but
for case B from 08:00 LST 1 July to 02:00 LST 2 July 2006 over R2 (denoted Fig. 12d).
**Figure 15** Same as Fig. 14 but for differences in the CON term (shaded; units: $10^{-4}$ g$^{-1}$kg$^{-1}$ s$^{-1}$)
and MA term (black lines; units: $10^{-4}$ g$^{-1}$ kg$^{-1}$ s$^{-1}$) in eq. (2).





772 **Figure 16** Same as Fig. 15 but for differences in horizontal wind speed (black lines; units: m s$^{-1}$)

773 and moisture flux (shade; units: $10^{-2}$ m kg kg$^{-1}$ s$^{-1}$).





**Table 1** Configurations of the WRF physics schemes used in the present study.

| Physical processes | Parameterization Scheme |
|---|---|
| Microphysics | Morrison 2-moment scheme (Morrison et al., 2009) |
| Long-wave radiation | RRTMG scheme (Iacono et al., 2008) |
| Short-wave radiation | RRTMG scheme |
| Surface layer | Monin-Obukhov scheme (Monin and Obukhov, 1954) |
| Land surface process | Noah land-surface model (Chen et al., 1996; Chen and Dudhia, 2001) |
| Planetary boundary layer process | Mellor-Yamada-Jajic TKE scheme (Mellor and Yamada, 1982; Janijic, 2001) |





**Table 2** Numerical experiments and corresponding urban land use and aerosol emissions.

| Experiment | Land-use category | Anthropogenic emissions |
|---|---|---|
| LU06E06 | 2006 | 2006 |
| LU70E06 | 1970 | 2006 |
| LU70E70 | 1970 | 1970 |



**Table 3** Analysis strategies for the investigation of urban land-use and/or aerosol effects.

| Difference | Mechanism |
|---|---|
| LU06E06- LU70E06 | Urban |
| LU70E06- LU70E70 | Aerosol |
| LU06E06- LU70E70 | Urban and aerosol |

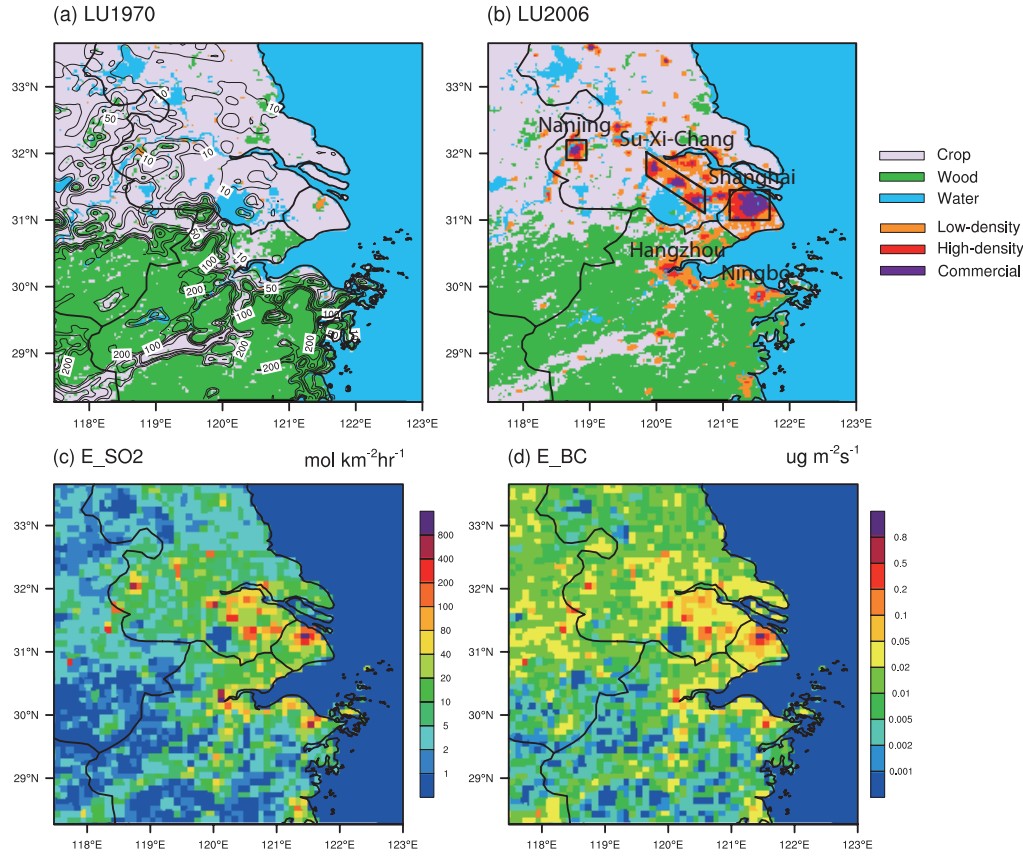

**Figure 1** Land-use categories for year (a) 1970; (b) 2006; and (c) $SO_2$ (units: mol km$^{-2}$ h$^{-1}$) and (d) black carbon (BC) emission rates (units: ug m$^{-2}$ s$^{-1}$) averaged over 2006-2010. The topography is also shown in Fig. 1a (contour; units: m). The boxes in Fig. 1b outline three mega-city clusters of Nanjing, Su-Xi-Chang, and Shanghai.





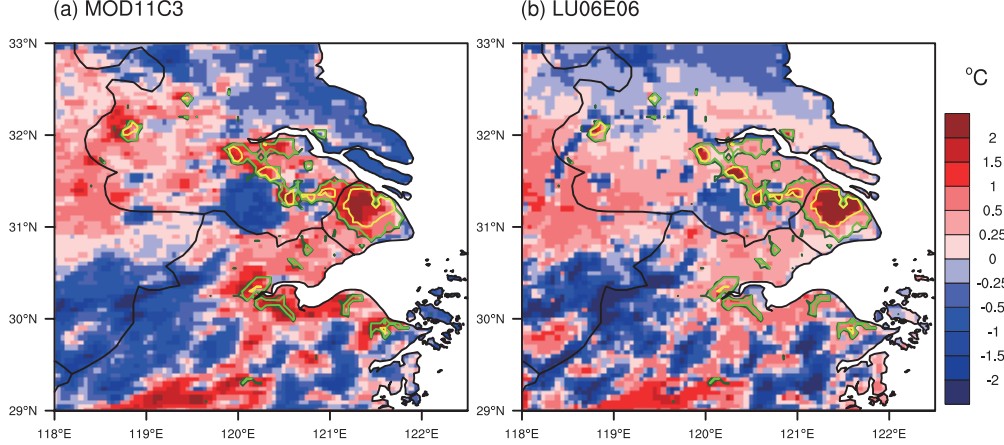

**Figure 2** Moving spatial anomalies of averaged surface skin temperature (units: °C) with a filtering window size of 1° ×1° for (a) MODIS observation and (b) the L06E06 simulation. The "High Intensity Residential" and "Commercial/Industrial/Transportation" areas are marked with green lines and yellow lines, respectively.





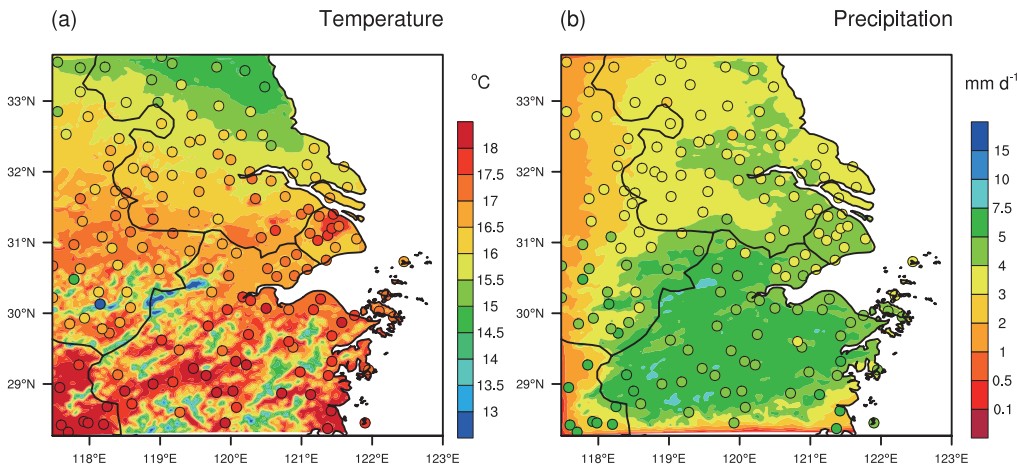

**Figure 3** Annual mean (a) near-surface temperature (units: °C) and (b) precipitation (unit: mm d$^{-1}$) from observations (shaded circles) and simulation of the LU06E06 (shaded).





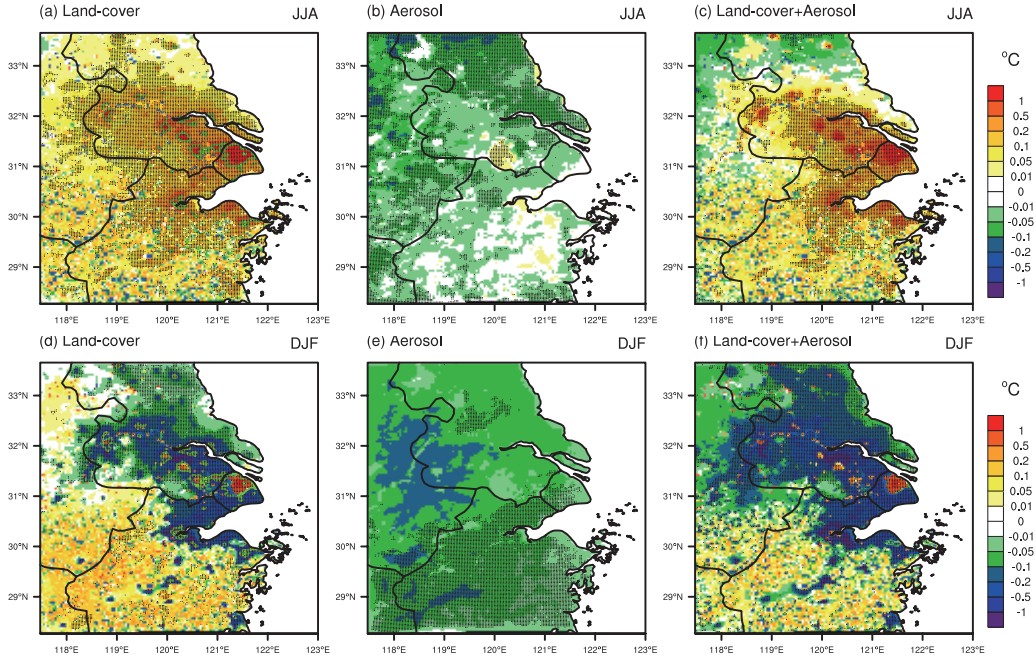

**Figure 4** Differences in mean 2-m temperature (Units: °C) between simulations (a, d) LU06E70 and LU70E70, (b, e) LU70E06 and LU70E70, (c, f) LU06E06 and LU70E70 for summer (upper panels) and winter (bottom panels). "Commercial/Industrial/Transportation" areas are marked with green lines. The black dots mark the area with statistically significant changes.



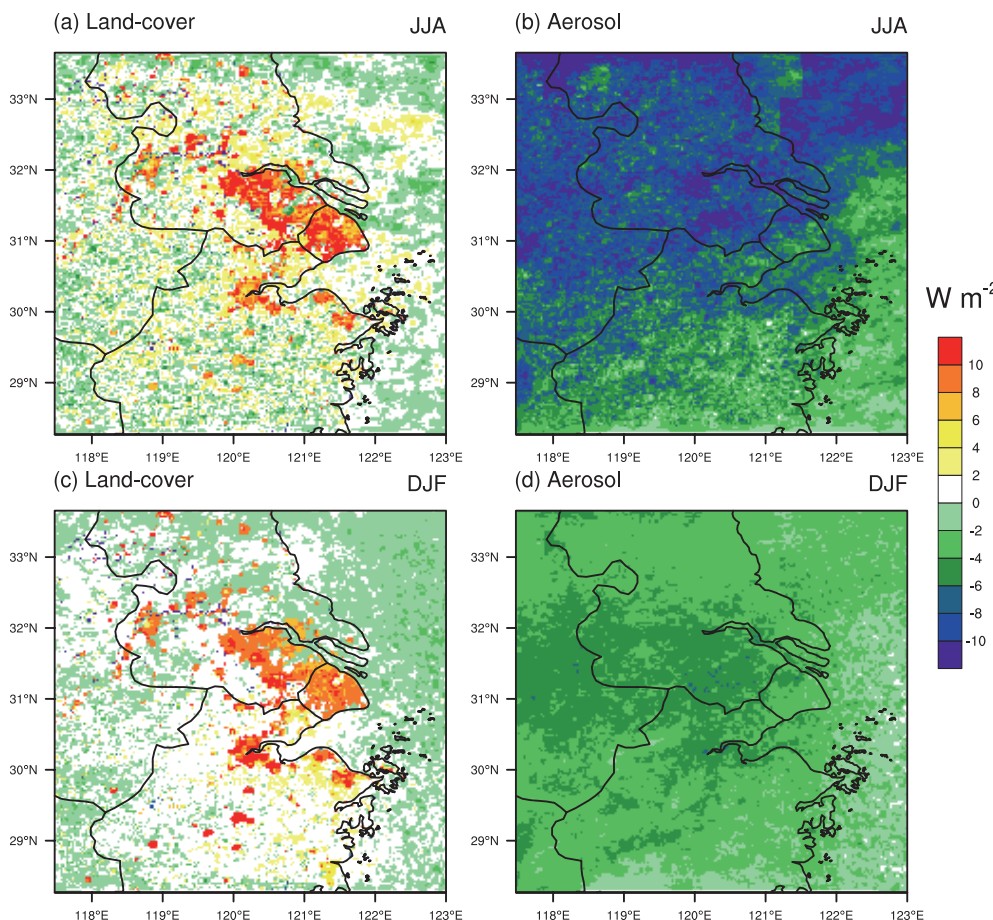

**Figure 5** Differences in net shortwave fluxes at the surface (units: W m$^{-2}$) between simulations (a, c) LU06E70 and LU70E70, and (b, d) LU70E06 and LU70E70 in summer (upper panels) and winter (bottom panels).





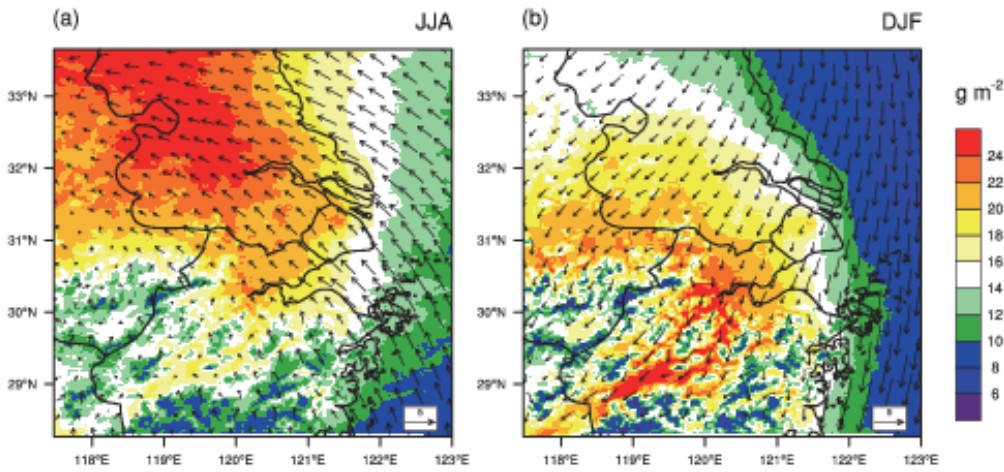

**Figure 6** Differences in column burden of PM2.5 (g m$^{-2}$) between simulations LU70E06 and

LU70E70, superimposed with near-surface winds simulated in LU70E70, for (a) summer and (b)

winter.



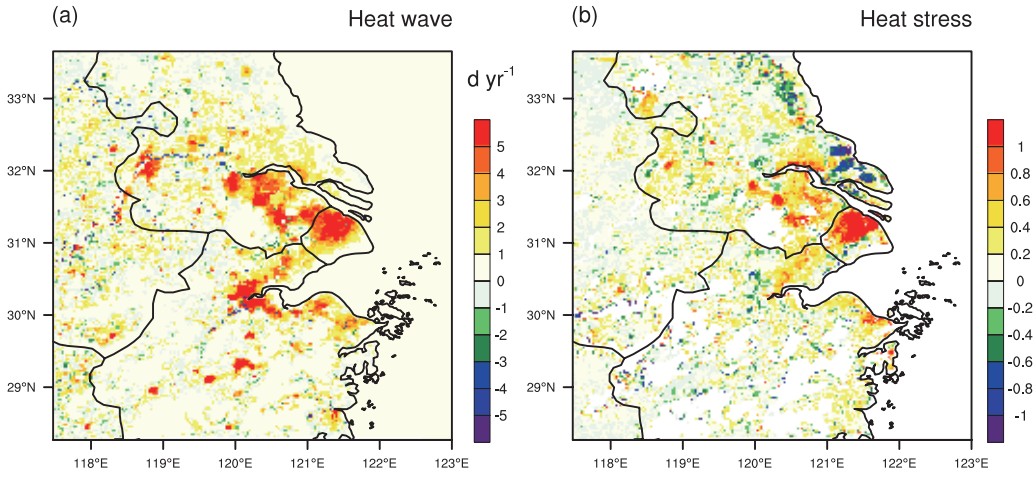

**Figure 7** Differences in mean summertime (a) heat wave days (units: d/yr) and (b) heat stress

(units: °C) between simulations LU06E70 and LU70E70.





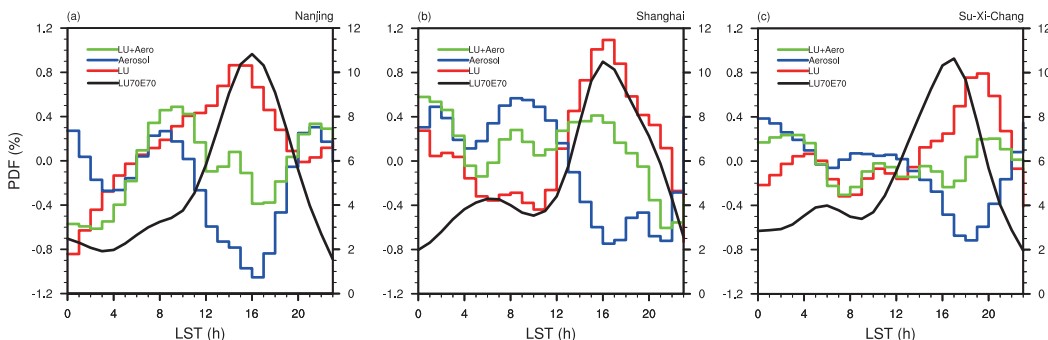

**Figure 8** Diurnal cycles of the frequency of summertime extreme rainfall events (defined using hourly precipitation intensity above 95[th] percentile, black lines, right axis) and the differences between simulations LU06E70 and LU70E70 (red lines), LU70E06 and LU70E70 (blue lines, left axis), and LU06E06 and LU70E70 (green lines) over (a) Nanjing, (b) Shanghai, and (c) Su-Xi-Chang.



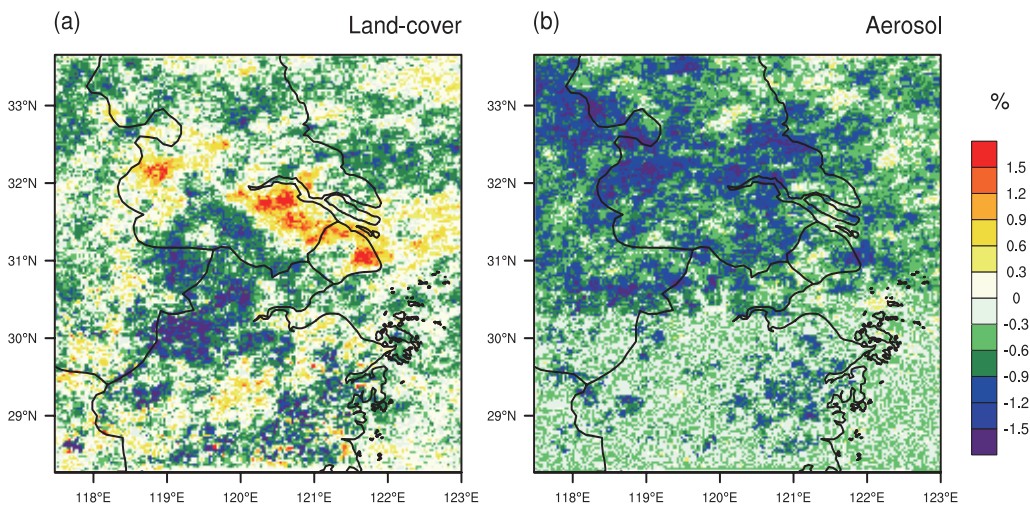

**Figure 9** Differences in the frequency of summertime extreme rainfall events (averaged from 12:00 to 20:00 LST) between simulations (a) LU06E70 and LU70E70, and (b) LU70E06 and LU70E70.




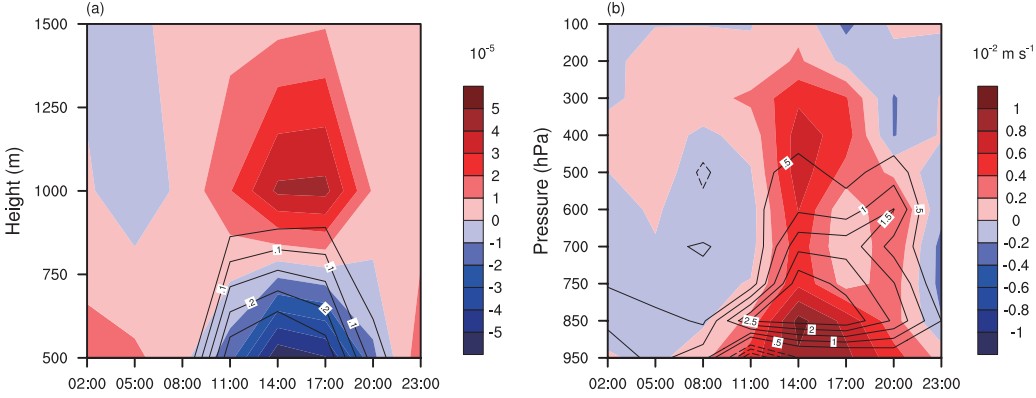

**Figure 10** (a) Time-height cross-sections of differences (between LU06E70 and LU70E70) in temperature (contour; units: °C) and divergence (shade; units: $10^{-5}$ s$^{-1}$) averaged over the three city clusters (Nanjing, Shanghai, and Su-Xi-Chang); (b) same as (a), but for vertical velocity (shade; units: $10^{-2}$ m s$^{-1}$) and cloud water mixing ratio (contour; $10^{-3}$ kg kg$^{-1}$).





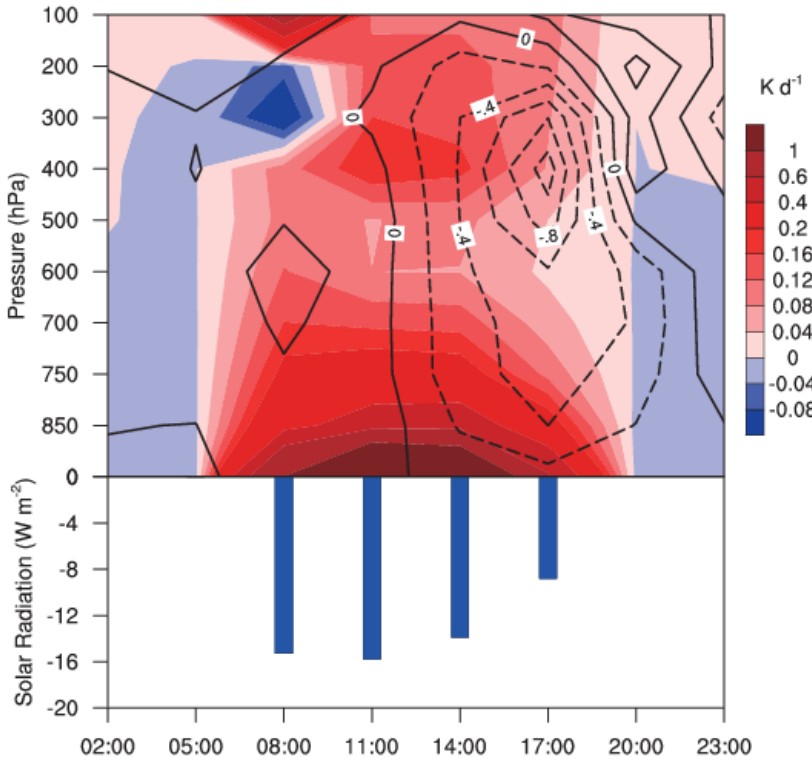

**Figure 11** Time-height cross-sections of differences (between LU70E06 and LU70E70) in radiative heating profile (shade; units: K d$^{-1}$), vertical velocity (contour; units: $10^{-2}$ m s$^{-1}$) and surface solar radiation (blue bars; units: W m$^{-2}$) averaged over the three city clusters (Nanjing, Shanghai, and Su-Xi-Chang).





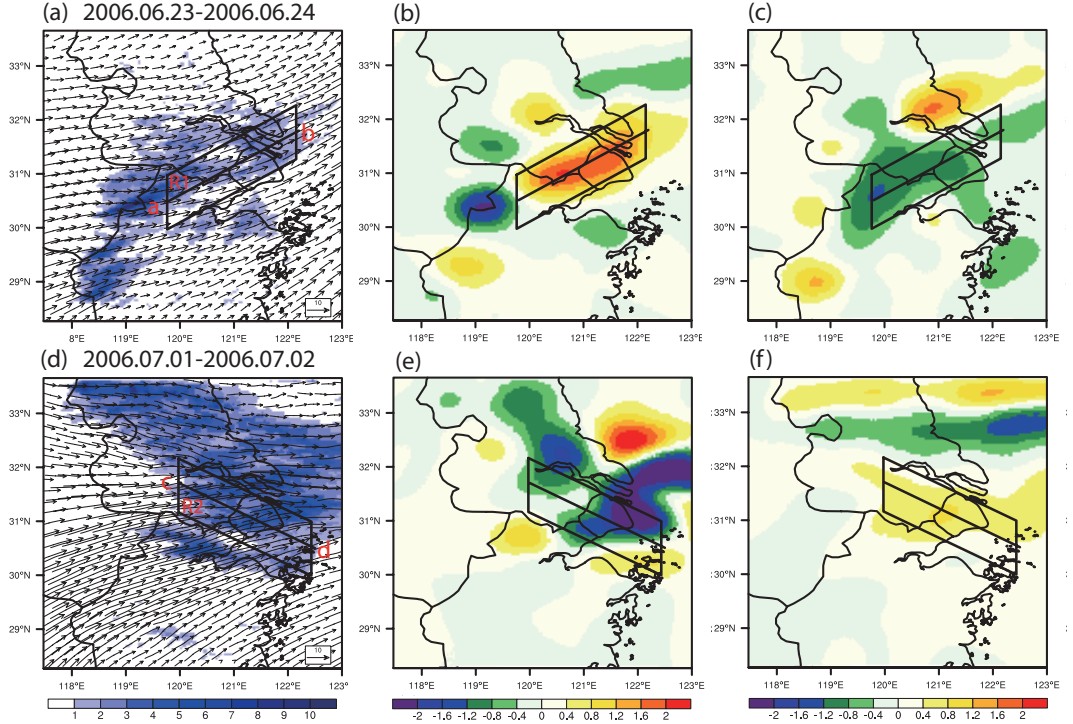

**Figure 12** Rain rate (units: mm h-1) superimposed with wind vectors at 850 hPa for case A from 08:00 LST 23 June to 08:00 LST 24 June 2006 (a) simulated in the LU06E06 simulation, (b) differences between LU06E70 and LU70E70, (c) differences between LU70E06 and LU70E70. Panels (d-f) are the same as (a-c) but for case B from 08:00 LST 1 July to 08:00 LST 2 July 2006. The boxes R1 in (a), R2 in (d) outline the three regions over which further analysis are conducted. Lines across the center of each box mark the cross-sections to be analyzed.





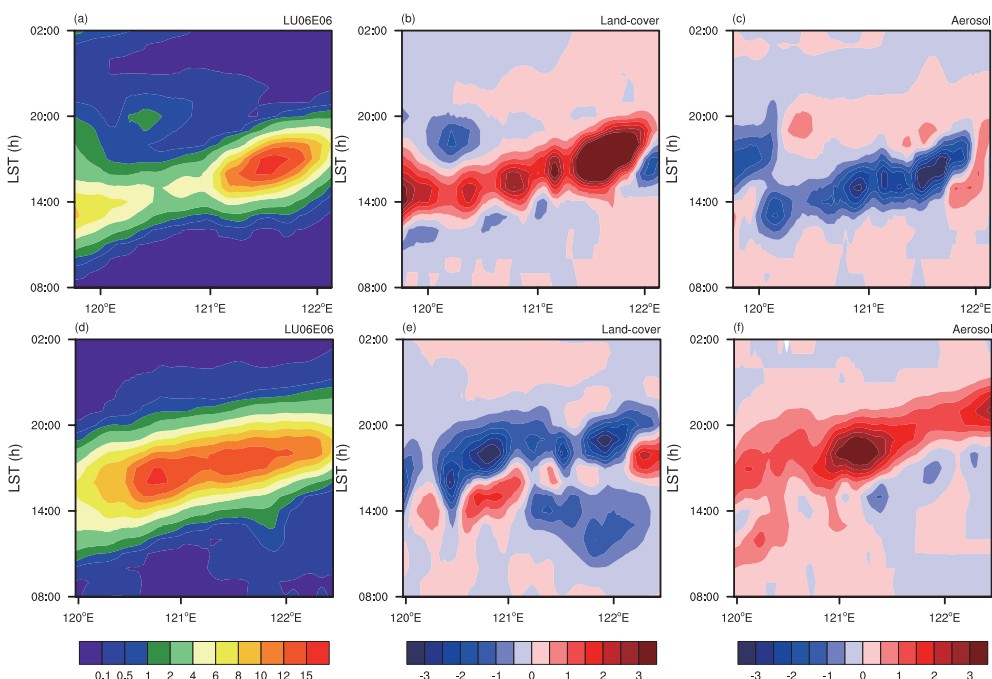

**Figure 13** The time evolution of precipitation (units: mm h$^{-1}$) along the line *ab* (marked in Fig. 12a) from 08:00 LST 23 June to 02:00 LST 24 June 2006 (case A) (a) simulated in the LU06E06 simulation, (b) differences between LU06E70 and LU70E70, (c) differences between LU70E06 and LU70E70. Panels (d-f) are the same as (a-c) but for case B along line *cd* (marked in Fig. 12d) from 08:00 LST 1 July to 02:00 LST 2 July 2006.

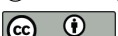



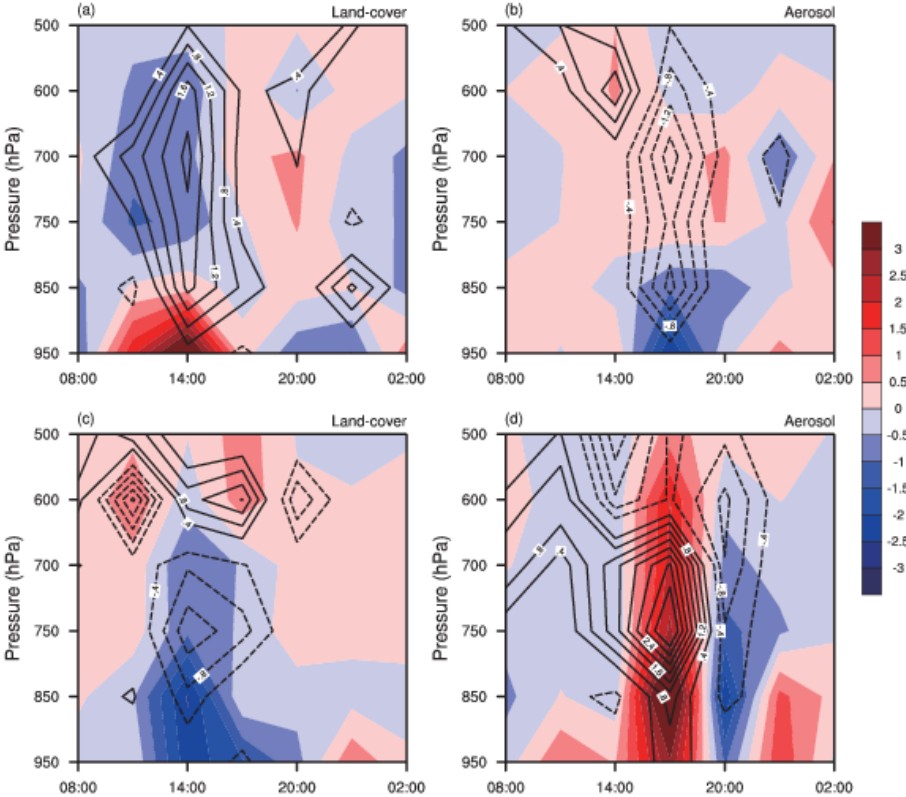

**Figure 14** The time-height cross-sections of differences in moisture flux convergence (shaded;

units: $10^{-4}$ g$^{-1}$ kg$^{-1}$ s$^{-1}$) and water vapor mixing ratio (black lines; units: $10^{-2}$ g kg$^{-1}$) from 08:00

LST 23 June to 02:00 LST 24 June 2006 (case A) over region R1 (denoted in Fig. 12a) between

(a) LU06E70 and LU70E70; (b) LU70E06 and LU70E70; Panels (c, d) are the same as (a, b) but

for case B from 08:00 LST 1 July to 02:00 LST 2 July 2006 over R2 (denoted Fig. 12d).





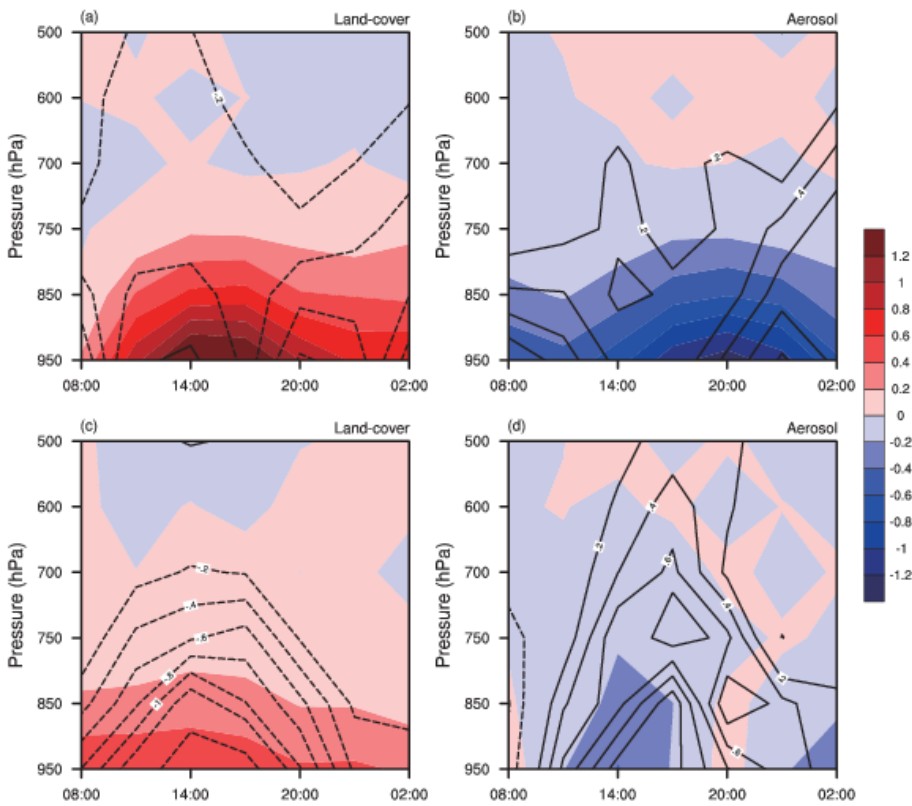

**Figure 15** Same as Fig. 14 but for differences in the CON term (shaded; units: $10^{-4}\,g^{-1}kg^{-1}\,s^{-1}$)

and MA term (black lines; units: $10^{-4}\,g^{-1}\,kg^{-1}\,s^{-1}$) in eq. (2).




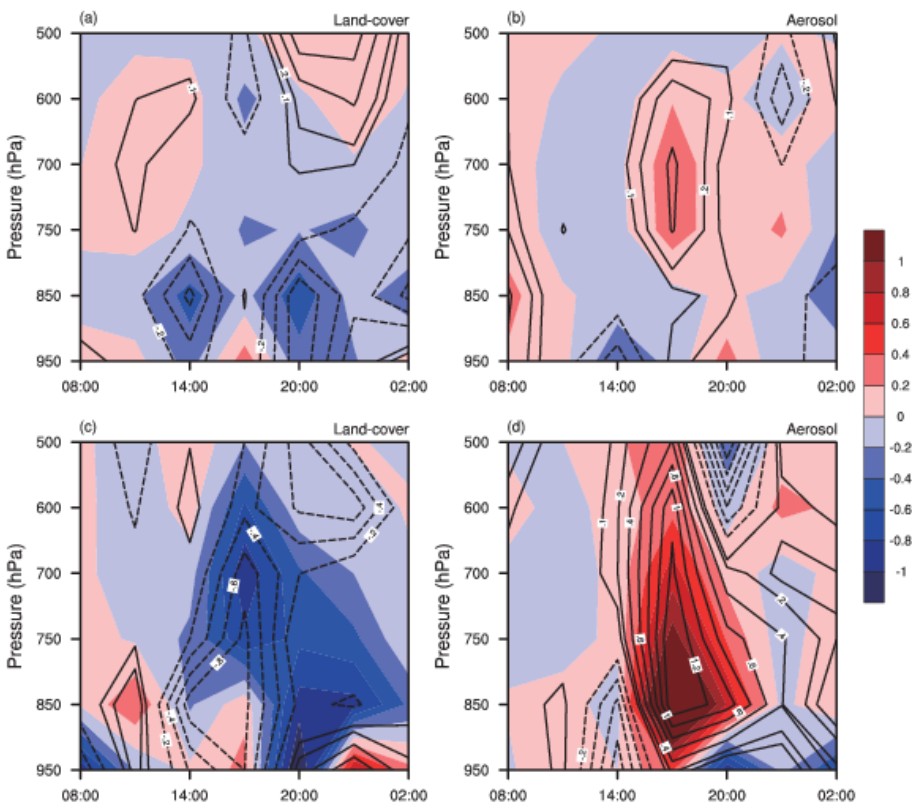

**Figure 16** Same as Fig. 15 but for differences in horizontal wind speed (black lines; units: m s-1)

and moisture flux (shade; units: 10-2 m kg kg-1 s-1). Noah only uses the dominant land cover

type, so no subgrid variability is simulated.