# Peer review of "Urbanization-induced urban heat island and aerosol effects on # climate extremes in the Yangtze River Delta Region of China"

_Atmospheric Chemistry and Physics, 2016_

## Referee Comment (RC1) · Anonymous Referee #2 · 19 Jan 2017

The authors performed fine-resolution model simulations using coupled WRF-Chem model to study the individual and combined effect of changes in land use and atmospheric aerosol loading from 1970 to 2006 on the climatic changes. The paper is generally well-written and the results are well discussed. I have following comments before it is accepted to be published on ACP.

One major comment: According to Table 2&3, the authors quantified the urban land use changes on the climatic effect under the atmospheric aerosol loading in 2006, and the aerosol loading changes on the climatic effect under the land use in 1970. Have the authors considered to run extra sensitivity simulation with LU06E70, to quantify these effects under the same year's base condition? How the authors consider the

uncertainties associated with that?

Minor comments: Pg 9: line 178: Can the authors elaborate why they consider the nighttime light correction for deriving the land use data in 2006 but not for 1970?

Pg 12: line 241-243, the description of the Figure 6 is not the same as in Pg 47. Please double check the imposed surface wind speed is from LU70E70 or LU06E06? Change "PM2.5" to "PM2.5"; Also update the quality of Figure 6. It is less clear compared with other figures.

Pg 12: line 252: have the authors consider how the different definition of the heatwave could affect the results?

Pg 44: figure 3, change "unit" to "units"

Pg 49: Figure 8: I would suggest the authors to rewrite the captions for Figure 8. Since it no longer shows the subtitle of "Land-Cover" "Aerosol" as in Figures 4 &5, I think it is better to express that the red lines are for Land cover, blue lines for Aerosol effect, and green lines for total.

Pg 51: Figure 10 (a), missing the units of "10-5 s-1" in the top of the vertical colorbar.

---

## Referee Comment (RC2) · Anonymous Referee #1 · 25 Feb 2017

Review comments on "Urbanization-induced urban heat heat island and aerosol effects on climate extremes in the Yangtze River Delta Region of China" by Zhong et al.

The influences of urbanization-induced land cover change and the aerosol concentrations on local and regional climate in the Yangtze River Delta in China were investigated by performing three sensitivity experiments using WRF-Chem model at convection-permitting scale (3km). Their separated and combined effects on precipitation and temperature were examined and compared. Moreover, the authors found the effects of external forcing were affected by the synoptic forcing. The manuscript is well written and contains some interesting results. Some comments are as follows.

1. Abstract L38-42: The role of synoptic forcing was not well summarized. These descriptions were too general.

2. One issue about the experimental design: the NCEP FNL reanalysis data with 1 degree was directly used to drive the WRF model at 3km. The ratio of the resolution of driving data to that of the regional climate model is about 40, which is quite large. The authors should justify this issue.

3. L152: how about the variation from 0800 to 1700? Linearly?

4. L198-L207: the authors evaluated the model performance in terms of the annual mean values. However, the changes of summer and winter climate were analyzed respectively in the following sections. So how about the model performance in simulating summer and winter climate?

5. Figure 6: the quality of this figure is poor due to its low resolution.

6. L403-L412: The authors stated that "the differences in the responses of moisture advection between two cases are related to different background circulation". I am not very convinced about this argument. In fact, the changes in moisture advection could be further decomposed into three terms, as shown below:

$$-\Delta\langle V \cdot \nabla q\rangle = -\langle V_{ctr} \cdot \Delta(\nabla q)\rangle - \langle(\nabla q)_{ctr} \cdot \Delta V\rangle - \langle\Delta(\nabla q) \cdot \Delta V\rangle$$

$\Delta$ () represents the difference between the sensitivity and control simulations, and the

subscript 'ctrl' denotes the control experiment. The first term in the right-hand side of is associated with the change in water vapor, while the second term is associated with the change in circulation. The third term is a nonlinear term including the contribution of both the moisture and circulation changes. This decomposition could answer whether the background circulation is indeed very important as the authors stated.

---

## Author Comment (AC1) · 29 Mar 2017

We thank both reviewers for their helpful comments and suggestions. In the attached PDF file "response_r2.docx" we explain how the comments are addressed and also attached the revised manuscript with changes tractable.

Please also note the supplement to this comment:
http://www.atmos-chem-phys-discuss.net/acp-2016-953/acp-2016-953-AC1-supplement.zip
* * *

---

## Author Comment (AC2) · 29 Mar 2017

We thank both reviewers for your helpful comments and suggestions. In the attached PDF file "response_r2.docx" we explain how the comments are addressed and also attached the revised manuscript with changes tractable.

Please also note the supplement to this comment:
http://www.atmos-chem-phys-discuss.net/acp-2016-953/acp-2016-953-AC2-supplement.zip
* * *

---

## Author Response (AR1)

We thank both reviewers for their helpful comments and suggestions. Below we explain how the comments are addressed and make note of changes in the revised manuscript.

**Reviewer #1:**

The authors performed fine-resolution model simulations using coupled WRF-Chem model to study the individual and combined effect of changes in land use and atmospheric aerosol loading from 1970 to 2006 on the climatic changes. The paper is generally well-written and the results are well discussed. I have following comments before it is accepted to be published on ACP.

One major comment:

1. According to Table 2&3, the authors quantified the urban land use changes on the climatic effect under the atmospheric aerosol loading in 2006, and the aerosol loading changes on the climatic effect under the land use in 1970. Have the authors considered to run extra sensitivity simulation with LU06E70, to quantify these effects under the same year's base condition? How the authors consider the uncertainties associated with that?

**Response:** We agree with the reviewer that we can also quantify the land cover effect under the aerosol loading in 1970 and the aerosol effect under the land use in 2006 by conducting an additional simulation LU06E70. We carefully thought about this during our experimental design, but we don't expect the land cover effect and the aerosol effect would substantially depend on the background aerosol condition and land use condition, respectively. On the other hand, the multi-year high-resolution WRF-Chem simulation is computationally expensive. It took nearly two wall-clock years to conduct the three experiments, so it wasn't feasible to conduct the additional LU06E70 simulation. The results of aerosol effects by using LU06E06–LU06E70 are likely to be quantitatively different from those using LU70E06–LU70E70, but we believe the differences will not change the qualitative results and major findings/conclusions in this study. The uncertainties in aerosol effect associated with this would not be even comparable to those in aerosol emissions and model physics.

Minor comments:

1. Pg 9: line 178: Can the authors elaborate why they consider the nighttime light correction for deriving the land use data in 2006 but not for 1970?

**Response:** We didn't use the correction for land use data for 1970 because the nighttime light data started from 1992. So we used the USGS data for 1970 instead. The urban area, which should be very small comparing with 2006, is ignored in 1970 USGS data.

2. Pg 12: line 241-243, the description of the Figure 6 is not the same as in Pg 47. Please double check the imposed surface wind speed is from LU70E70 or LU06E06? Change "PM$_{2.5}$" to "PM2.5"; Also update the quality of Figure 6. It is less clear compared with other figures.

**Response:** Thanks for the reviewer's good catch. The imposed surface wind speed is indeed from LU70E70. This has now been corrected in the description of Fig. 6 in the revised manuscript (Line 248). PM$_{2.5}$ has also been changed to PM2.5 as suggested. The quality of Figure 6 has also been much improved in the revised manuscript.

3. Pg 12: line 252: have the authors consider how the different definition of the heatwave could affect the results?

**Response:** Yes, we did also consider other definitions of heatwave. For instance, according to the definition by WMO, heatwave occurs when the temperature reaches or exceeds 32°C for 3 consecutive days or more. In this study, the definition we choose is according to the China Meteorological Administration (CMA), which is believed to be more suitable for the regional climate in the Yangtze River Delta Region. Figure R1 illustrates the land cover effect on heatwave days for the two definitions respectively. We can see that the spatial patterns of results for both definitions are quite similar, but there an obvious increase in heatwave days over the major mega cities when the temperature threshold for heatwaves is lower (32°C vs. 35°C) as expected. Although the increase in heatwave is greater for the WMO definition, with an average rate of 8.7 d/yr in the major mega cities, the qualitative conclusion doesn't change.

[Figure]

**Figure R1** Differences in mean summertime heatwave days (units: d/yr) between LU06E70 and LU70E70 for (a) CMA and (b) WMO definition.

4. Pg 44: figure 3, change "unit" to "units"

**Response:** Changed it in the revised manuscript.

5. Pg 49: Figure 8: I would suggest the authors to rewrite the captions for Figure 8. Since it no longer shows the subtitle of "Land-Cover" "Aerosol" as in Figures 4 &5, I

think it is better to express that the red lines are for Land cover, blue lines for Aerosol effect, and green lines for total.

**Response:** We have made the suggested change for clarity.

6. Pg 51: Figure 10 (a), missing the units of "10-5 s-1" in the top of the vertical colorbar.

**Response:** Thanks for catching that. The units have been added as suggested.

**Reviewer #2:**

1. Abstract L38-42: The role of synoptic forcing was not well summarized. These descriptions were too general.

**Response:** Thanks for the suggestion. In the revised manuscript, we have added a more detailed summary for the role of synoptic forcing (Line 40-46).

2. One issue about the experimental design: the NCEP FNL reanalysis data with 1 degree was directly used to drive the WRF model at 3km. The ratio of the resolution of driving data to that of the regional climate model is about 40, which is quite large. The authors should justify this issue.

**Response:** The lateral boundary condition is provided to the WRF domain using the NCEP reanalysis data via linear interpolation, so it can represent the horizontal linear variation of meteorological data, no matter how much the ratio of the resolution is. In previous studies that use a regional model to do similar long-term simulations, the NCEP data was also directly used to drive the model at the resolution of less than 10km (e. g. Wang et al., 2015). More importantly, in our model evaluation section, it is shown that the model can generally capture the annual mean climate in the domain.

Wang, X. M., Sun, X. G., Tang, J. P., and Yang, X. Q.: Urbanization-induced regional warming in Yangtze River Delta: potential role of anthropogenic heat release, Int. J. Climatol., doi: 10.1002/joc.4296, 2015.

3. L152: how about the variation from 0800 to 1700? Linearly?

**Response:** No, it is non-linear. The figure below shows the default diurnal variation of AH in WRF.

[Figure]

**Figure R2** Diurnal cycle of anthropogenic heating (AH) normalized by the peak value of 50 W m-2 for the WRF default.

4. L198-L207: the authors evaluated the model performance in terms of the annual mean values. However, the changes of summer and winter climate were analyzed respectively in the following sections. So how about the model performance in simulating summer and winter climate?

**Response:** We have also evaluated the model performance in terms of both summer and winter climate. Fig. R3 illustrates the averaged near-surface temperature and precipitation in summer and winter respectively. The simulated spatial pattern of near-surface air temperature agrees well with observations for both summer and winter, with high temperature centers located at meteorological stations in major cities. The model generally captures the observed precipitation except for the overestimation in summer and the underestimation in winter over the southern part of the domain.

[Figure]

**Figure R2** (a) near-surface temperature (units: °C) and (b) precipitation (units: mm d$^{-1}$) from observations (shaded circles) and simulation of the LU06E06 (shaded) for summer. Panels (c-d) are the same as (a-b) but for winter.

5. Figure 6: the quality of this figure is poor due to its low resolution.

**Response:** The quality of this figure has been much improved it in the revised manuscript.

6. L403-L412: The authors stated that "the differences in the responses of moisture advection between two cases are related to different background circulation". I am not very convinced about this argument. In fact, the changes in moisture advection could be further decomposed into three terms, as shown below:

$$-\Delta V \cdot \nabla q = -V \text{ ctrl} \cdot \Delta(\nabla q) - (\nabla q) \text{ ctrl} \cdot \Delta V - \Delta(\nabla q) \cdot \Delta V$$

$\Delta$ () represents the difference between the sensitivity and control simulations, and the subscript 'ctrl' denotes the control experiment. The first term in the right-hand side of is associated with the change in water vapor, while the second term is associated with the change in circulation. The third term is a nonlinear term including the contribution of both the moisture and circulation changes. This decomposition could answer whether the background circulation is indeed very important as the authors stated.

**Response:** Thanks for the suggestion. We have calculated these three terms in the decomposition as suggested. Fig. R3 illustrates time-height cross section of the changes in the first and the second term, respectively. The contribution of the third nonlinear term is small and negligible compared to the other two terms (figure not shown). We can see that the most significant difference between these two cases is the change in the first term, which is directly associated with the background circulation. Therefore, the changes in moisture advection (MA) are much larger in case B due to the stronger background winds. We have used this figure to replace the original Fig. 16 to make our argument more convincing.

[revised manuscript text omitted]